# Perinatal deaths attributable to congenital heart defects in Hunan Province, China, 2016–2020

Xu Zhou[1]*, Jian He[1], Haiyan Kuang[1]*, Junqun Fang[1]*, Hua Wang[2,3]*

**1** Hunan Provincial Maternal and Child Health Care Hospital, Changsha, Hunan Province, China, **2** The Hunan Children's Hospital, Changsha, Hunan Province, China, **3** National Health Commission Key Laboratory of Birth Defects Research, Prevention and Treatment, Hunan Provincial Maternal and Child Health Care Hospital, Changsha, Hunan, China

* 15343689@qq.com (HK); 40112079@qq.com (JF); wanghua213@aliyun.com (HW); chzhouxu@163.com (XZ)

**Data Availability Statement:** The analysis code deposited at time of publication is available from the OSF repository (https://doi.org/10.17605/OSF.IO/VEAM8).

## Abstract

### Objective

To explore the association between demographic characteristics and perinatal deaths attributable to **congenital heart defects (CHDs)**.

### Methods

Data were obtained from the Birth Defects Surveillance System of Hunan Province, China, 2016–2020. The surveillance population included fetuses and infants from 28 weeks of gestation to 7 days after birth whose mothers delivered in the surveillance hospitals. Surveillance data included demographic characteristics such as sex, residence, maternal age, and other key information, and were used to calculate the prevalence of CHDs and perinatal mortality rates (PMR) with 95% confidence intervals (CI). Multivariable logistic regression analysis (method: Forward, Wald, α = 0.05) and adjusted odds ratios (ORs) were used to identify factors associated with perinatal deaths attributable to CHDs.

### Results

This study included 847755 fetuses, and 4161 CHDs were identified, with a prevalence of 0.49% (95%CI: 0.48–0.51). A total of 976 perinatal deaths attributable to CHDs were identified, including 16 (1.64%) early neonatal deaths and 960 (98.36%) stillbirths, with a PMR of 23.46% (95%CI: 21.98–24.93). In stepwise logistic regression analysis, perinatal deaths attributable to CHDs were more common in rural areas than urban areas (OR = 2.21, 95% CI: 1.76–2.78), more common in maternal age <20 years (OR = 2.40, 95%CI: 1.05–5.47), 20–24 years (OR = 2.13, 95%CI: 1.46–3.11) than maternal age of 25–29 years, more common in 2 (OR = 1.60, 95%CI: 1.18–2.18) or 3 (OR = 1.43, 95%CI: 1.01–2.02) or 4 (OR = 1.84, 95%CI: 1.21–2.78) or >= 5 (OR = 2.02, 95%CI: 1.28–3.18) previous pregnancies than the first pregnancy, and more common in CHDs diagnosed in >= 37 gestational weeks (OR = 77.37, 95%CI: 41.37–144.67) or 33–36 gestational weeks (OR = 305.63, 95%CI: 172.61–

**Funding:** This research was financially supported by the Hunan Provincial Health Commission in the form of a grant (20230883) received by HK. No additional external funding was received for this study.

**Competing interests:** The authors have declared that no competing interests exist.

541.15) or < = 32 gestional weeks (OR = 395.69, 95%CI: 233.23–671.33) than diagnosed in postnatal period (within 7 days), and less common in multiple births than singletons (OR = 0.48, 95%CI: 0.28–0.80).

## Conclusions

Perinatal deaths were common in CHDs in Hunan in 2016–2020. Several demographic characteristics were associated with perinatal deaths attributable to CHDs, which may be summarized mainly as economic and medical conditions, severity of CHDs, and parental attitudes toward CHDs.

## Introduction

Perinatal deaths include stillbirths and early neonatal deaths from 28 weeks of gestation to 7 days after birth [1]. The **perinatal mortality rate (PMR)** PMR is the number of perinatal deaths per 100 fetuses. Birth defects are structural or functional anomalies at or before birth [2]. The global prevalence of birth defects is approximately 2% -3% [3]. **Congenital heart defect (CHD)** is typically defined as structural anomalies of the heart or great vessels at or before birth [4]. CHD is the most common birth defect, accounting for nearly 1/3 of all major birth defects [5, 6]. The global birth prevalence of CHDs is 0.94% (2010–2017) [4]. Severe birth defects, including many severe CHDs, significantly increase the risk of perinatal deaths [7–9]. In developed countries such as Europe and the United States, birth defects have long been the leading cause of perinatal deaths [10]. The World Health Organization estimated that approximately 12.6% of neonatal deaths worldwide are related to birth defects each year [11].

There are several studies on perinatal deaths attributable to birth defects or several specific defects (e.g., gastroschisis, omphalocele, cleft lip, and palate) [12–18]. However, to our knowledge, there are relatively few studies on perinatal deaths attributable to CHDs, and there are limitations in previous studies. For example, relatively few studies have reported the PMR of CHDs. In 2023, we conducted a study on the PMR of a broad range of specific defects (including total CHDs) in Hunan Province, China (2010–2020), and the PMR of total CHDs was 23.15%. However, in this study, we did not report the PMR of specific CHDs, and they did not explore the association between demographic characteristics and perinatal deaths using multivariable logistic regression analysis, as pointed out in the discussion section [9]. Zhang et al. found that 17.1% of stillbirths and 1.2% of early neonatal deaths in Zhejiang Province, eastern China (2014–2018) were associated with CHDs [6]. Dolk et al. reported a PMR of 3.6% for non-chromosomal CHDs in Europe (2000–2005) [19]. There are significant differences between these studies.

Relatively few studies have been conducted on perinatal deaths attributable to a wide range of specific CHDs. Beroukhim et al. reported a PMR of approximately 40% for single-ventricle cardiac defects at Boston Children's Hospital, Boston, Massachusetts, United States (1995–2008) [20], and this investigation was conducted at a single hospital earlier. In addition, relatively few studies have been conducted on the factors associated with perinatal deaths attributable to CHDs. Pace et al. examined the first-year survival of infants with CHDs. They investigated the potential role of socio-economic and demographic factors on survival [21], but some factors (e.g., maternal age and residence) were not included. To our knowledge, the association between demographic characteristics and perinatal deaths attributable to CHDs has not been investigated using multivariable logistic regression analysis.

Therefore, we conducted a multivariable logistic regression analysis based on data from the Birth Defects Surveillance System (2016–2020) of Hunan Province in south-central China to explore the association between demographic characteristics and perinatal deaths attributable to CHDs.

## Methods

### Data sources

Data were obtained from the Birth Defects Surveillance System of Hunan Province, China (2016–2020). The Hunan Provincial Health Commission manages the system, which has selected 52 representative registered hospitals as hospital-based birth defects surveillance sites since 1996. In order to make the surveillance data representative, the Hunan Provincial Health Commission conducted an expert assessment before selecting the surveillance sites. These 52 hospitals are appropriately distributed throughout the province. They are the main delivery hospital in their regions, with about 150,000 to 200,000 live births annually, accounting for 1/4 to 1/3 of all live births in Hunan Province.

The surveillance population included all live births, stillbirths, infant deaths, and legal termination of pregnancy from 28 weeks of gestation to 7 days after birth that occurred in the surveillance site. The surveillance data included demographic characteristics such as sex, residence, maternal age, and other key information. Eight demographic characteristics that may be associated with perinatal deaths attributable to CHDs were selected for analysis in this study, including sex, residence, maternal age, maternal education level, number of births, number of pregnancies, parity, and time of diagnosis. According to the Hunan Maternal and Child Health Surveillance Manual, obstetricians or neonatologists are responsible for collecting surveillance data. Birth defects were coded according to the International Classification of Diseases (10th Revision, codes Q00–Q99). The ICD codes CHDs as Q20-Q26.

### Definitions

Perinatal deaths include stillbirths and early neonatal deaths from 28 weeks of gestation to 7 days after birth. The prevalence of birth defects is the number of birth defects per 100 fetuses (births and deaths at 28 weeks of gestation and later). PMR is the number of perinatal deaths per 100 fetuses.

### Ethics approval and consent to participate

The Hunan Provincial Health Commission has developed the Hunan Maternal and Child Health Surveillance Manual to routinely collect surveillance data. Therefore, there is no additional written informed consent. The Medical Ethics Committee of Hunan Provincial Maternal and Child Health Care Hospital approved the study. (NO: 2022-S64). It is a retrospective study of medical records, and we confirmed that all operations were following relevant guidelines and regulations.

### Data quality control

To carry out surveillance, the Hunan Provincial Health Commission formulated the Hunan Maternal and Child Health Surveillance Manual. Data were collected and reported by experienced doctors. To reduce surveillance data's integrity and information error rates, the Hunan Provincial Health Commission asked the technical guidance departments to conduct comprehensive quality control once a year.

## Statistical analysis

Prevalence and PMR of specific CHDs with 95% confidence intervals (CI) were calculated by the log-binomial method [22]. Chi-square trend tests ($\chi^2_{trend}$) were used to determine trends in prevalence and PMR by year, maternal age, number of pregnancies, and time of diagnosis. Univariate analysis and unadjusted odds ratios (ORs) were calculated to examine the association of each demographic characteristic with perinatal deaths attributable to CHDs. Multivariable logistic regression analysis (method: Forward, Wald, α = 0.05) and adjusted ORs were used to identify factors associated with perinatal deaths attributable to CHDs. We used the presence or absence of perinatal deaths attributable to CHDs as the dependent variable, and the variables assessed significantly in univariate analysis, or that some studies have suggested were highly probable to be associated with CHDs, were entered as independent variables in multivariable logistic regression analysis.

Statistical analyses were performed using SPSS 18.0 (SPSS Inc. in Chicago, USA).

## Results

### Prevalence and PMR of CHDs

This study included 847755 fetuses. A total of 14459 birth defects were identified, including 4161 (28.78%) CHDs. The prevalence of CHDs was 0.49% (95%CI: 0.48–0.51). A total of 7445 perinatal deaths were identified, of which 3049 (40.95%) were attributable to birth defects and 976 (13.11%) were attributable to CHDs. Of the 976 perinatal deaths attributable to CHDs, 16 (1.64%) were early neonatal deaths, and 960 (98.36%) were stillbirths. 22.49% (936 cases) of CHDs ended in medical **termination of pregnancy (TOP)**. The overall PMR was 0.88 (95% CI: 0.86–0.90), and the PMRs of birth defects and CHDs were 21.09% (95%CI: 20.34–21.84) and 23.46% (95%CI: 21.98–24.93), respectively. From 2016 to 2020, the PMRs of CHDs were 28.55%, 20.85%, 26.57%, 22.01%, and 19.94%, respectively, showing a downward trend ($\chi^2_{trend}$ = 9.86, P = 0.002) (**Table 1**).

**Table 1. Prevalence and PMR of CHDs in Hunan Province, China, 2016–2020.**

| Year | Total fetuses | Total birth defects | | Total CHDs | | Total perinatal deaths | | Perinatal deaths attributable to birth defects | | | Perinatal deaths attributable to CHDs | | |
|------|-------|-----|----------------------|-----|----------------------|-----|-------------------|-----|----------------------|--------------------------------|-----|-------------------|----------------------------|
| | | n | Prevalence (%, 95%CI) | n | Prevalence (%, 95%CI) | n | PMR (%, 95%CI) | n | PMR of birth defects (%) | Among total perinatal deaths (%) | n | PMR of CHDs (%) | Among total perinatal deaths (%) |
| 2016 | 170688 | 3107 | 1.82(1.76–1.88) | 823 | 0.48(0.45–0.52) | 1901 | 1.11(1.06–1.16) | 744 | 23.95(22.23–25.67) | 39.14 | 235 | 28.55(24.90–32.20) | 12.36 |
| 2017 | 196316 | 3533 | 1.80(1.74–1.86) | 1055 | 0.54(0.50–0.57) | 1694 | 0.86(0.82–0.90) | 686 | 19.42(17.96–20.87) | 40.50 | 220 | 20.85(18.10–23.61) | 12.99 |
| 2018 | 177762 | 2900 | 1.63(1.57–1.69) | 734 | 0.41(0.38–0.44) | 1435 | 0.81(0.77–0.85) | 607 | 20.93(19.27–22.60) | 42.30 | 195 | 26.57(22.84–30.30) | 13.59 |
| 2019 | 164840 | 2643 | 1.60(1.54–1.66) | 827 | 0.50(0.47–0.54) | 1293 | 0.78(0.74–0.83) | 534 | 20.20(18.49–21.92) | 41.30 | 182 | 22.01(18.81–25.20) | 14.08 |
| 2020 | 138149 | 2276 | 1.65(1.58–1.72) | 722 | 0.52(0.48–0.56) | 1122 | 0.81(0.76–0.86) | 478 | 21.00(19.12–22.88) | 42.60 | 144 | 19.94(16.69–23.20) | 12.83 |
| Total | 847755 | 14459 | 1.71(1.68–1.73) | 4161 | 0.49(0.48–0.51) | 7445 | 0.88(0.86–0.90) | 3049 | 21.09(20.34–21.84) | 40.95 | 976 | 23.46(21.98–24.93) | 13.11 |

**Abbreviation**: PMR = perinatal mortality rate; CHD = congenital heart defect; CI = confidence intervals

## Prevalence and PMR of specific CHDs

More than half of fetuses with CHD had multiple specific CHDs (52.15%, 2170 / 4161), and 19.83% (825 / 4161) of CHDs were combined with other defects. Most perinatal deaths attributable to CHDs had congenital malformations of cardiac septa (689 cases, 70.59%), which mainly included ventricular septal defect (328 cases, 33.61%). Fetuses with congenital malformations of cardiac chambers and connections were more likely to end in perinatal deaths (PMR = 66.18%), followed by congenital malformations of aortic and mitral valves (PMR = 61.18%). Fetuses with the following specific CHDs were more likely to end in perinatal deaths: congenital mitral stenosis (80.00%), common arterial trunk (78.95%), double inlet ventricle (73.33%), stenosis of aorta (73.02%), hypoplastic left heart syndrome (72.73%), Tetralogy of Fallot (67.35%), double outlet right ventricle (67.24%), congenital stenosis of aortic valve (65.38%), hypoplastic right heart syndrome (64.29%), and stenosis of pulmonary artery (63.64%). **Table 2** shows the details of the PMR of specific CHDs (**Table 2**).

## Results of the univariate analysis

The PMR of CHDs was higher in rural than urban areas (35.62% vs. 14.01%, OR = 3.40, 95% CI: 2.92–3.95), while lower in multiple births than singletons (10.58% vs. 24.50%, OR = 0.36, 95%CI: 0.25–0.53). Compared to senior high school as maternal education level (PMR = 25.71%), PMR was higher in junior high school and below (PMR = 34.24%, OR = 1.50, 95%CI: 1.23–1.85), while lower in university and above (PMR = 18.63%, OR = 0.66, 95%CI: 0.56–0.78).

PMR of CHDs was negatively associated with maternal age ($\chi^2_{trend}$ = 52.33, P <0.001) and time of diagnosis ($\chi^2_{trend}$ = 2280.77, P <0.001), and was positively associated with number of pregnancies ($\chi^2_{trend}$ = 10.17, P = 0.001) (**Table 3**).

## Results of the multivariable logistic regression analysis

Starting with all variables from Table 3, the following variables were selected for the final model in stepwise logistic regression analysis: residence, maternal age, number of births, number of pregnancies, and time of diagnosis. In the logistic regression model with mutual adjustment, perinatal deaths attributable to CHDs were more common in rural areas than urban areas (OR = 2.21, 95%CI: 1.76–2.78) and less common in multiple births than in singletons (OR = 0.48, 95%CI: 0.28–0.80). Compared to maternal age 25–29 years, perinatal deaths attributable to CHDs were more common in <20 years (OR = 2.40, 95%CI: 1.05–5.47) or 20–24 years (OR = 2.13, 95%CI: 1.46–3.11), and less common in 30–34 years (OR = 0.75, 95%CI: 0.56–0.99) or > = 35 years (OR = 0.66, 95%CI: 0.46–0.96). Compared to the first pregnancy, perinatal deaths attributable to CHDs were more common in the number of pregnancies was 2 (OR = 1.60, 95%CI: 1.18–2.18) or 3 (OR = 1.43, 95%CI: 1.01–2.02) or 4 (OR = 1.84, 95%CI: 1.21–2.78) or > = 5 (OR = 2.02, 95%CI: 1.28–3.18). Compared to CHDs diagnosed in the postnatal period (within 7 days), perinatal deaths attributable to CHDs were more common in diagnosed in > = 37 gestational weeks (OR = 77.37, 95%CI: 41.37–144.67) or 33–36 gestational weeks (OR = 305.63, 95%CI: 172.61–541.15) or < = 32 gestational weeks (OR = 395.69, 95% CI: 233.23–671.33) (**Table 4**).

## Discussion

Overall, perinatal deaths were common in CHDs. Perinatal deaths attributable to CHDs were more common in rural areas, with low maternal age, high number of pregnancies, and diagnosis in the prenatal period (especially at low gestational age). To our knowledge, this study is the

**Table 2. Prevalence and PMR of specific CHDs.**

| Types | ICD codes | n | Prevalence (%, 95% CI) | Perinatal deaths (n) | PMR of specific CHDs (%) * |
|---|---|---|---|---|---|
| **Congenital malformations of cardiac chambers and connections** | **Q20** | **136** | **0.016(0.013–0.019)** | **90** | **66.18** |
| Common arterial trunk | Q20.0 | 19 | 0.002(0.001–0.003) | 15 | 78.95 |
| Double outlet right ventricle | Q20.1 | 58 | 0.007(0.005–0.009) | 39 | 67.24 |
| Discordant ventriculoarterial connection | Q20.3 | 30 | 0.004(0.002–0.005) | 17 | 56.67 |
| Double inlet ventricle | Q20.4 | 15 | 0.002(0.001–0.003) | 11 | 73.33 |
| Other | Other codes of Q20 | 20 | 0.002(0.001–0.003) | 14 | 70.00 |
| **Congenital malformations of cardiac septa** | **Q21** | **3660** | **0.432(0.418–0.446)** | **689** | **18.83** |
| Ventricular septal defect | Q21.0 | 1604 | 0.189(0.180–0.198) | 328 | 20.45 |
| Atrial septal defect | Q21.1 | 2327 | 0.274(0.263–0.286) | 30 | 1.29 |
| Atrioventricular septal defect | Q21.2 | 102 | 0.012(0.010–0.014) | 39 | 38.24 |
| Tetralogy of Fallot | Q21.3 | 147 | 0.017(0.015–0.020) | 99 | 67.35 |
| Other | Other codes of Q21 | 274 | 0.032(0.028–0.036) | 206 | 75.18 |
| **Congenital malformations of pulmonary and tricuspid valves** | **Q22** | **159** | **0.019(0.016–0.022)** | **70** | **44.03** |
| Congenital pulmonary valve stenosis | Q22.1 | 64 | 0.008(0.006–0.009) | 33 | 51.56 |
| Congenital pulmonary valve insufficiency | Q22.2 | 16 | 0.002(0.001–0.003) | 5 | 31.25 |
| Hypoplastic right heart syndrome | Q22.6 | 14 | 0.002(0.001–0.003) | 9 | 64.29 |
| Other congenital malformations of the tricuspid valve | Q22.8 | 53 | 0.006(0.005–0.008) | 14 | 26.42 |
| Other | Other codes of Q22 | 25 | 0.003(0.002–0.004) | 16 | 64.00 |
| **Congenital malformations of aortic and mitral valves** | **Q23** | **85** | **0.010(0.008–0.012)** | **52** | **61.18** |
| Congenital stenosis of aortic valve | Q23.0 | 26 | 0.003(0.002–0.004) | 17 | 65.38 |
| Congenital mitral stenosis | Q23.2 | 10 | 0.001(0.000–0.002) | 8 | 80.00 |
| Hypoplastic left heart syndrome | Q23.4 | 33 | 0.004(0.003–0.005) | 24 | 72.73 |
| Other | Other codes of Q23 | 22 | 0.003(0.002–0.004) | 8 | 36.36 |
| **Other congenital malformations of the heart** | **Q24** | **142** | **0.017(0.014–0.020)** | **72** | **50.70** |
| Dextrocardia | Q24.0 | 33 | 0.004(0.003–0.005) | 16 | 48.48 |
| Malformation of coronary vessels | Q24.5 | 23 | 0.003(0.002–0.004) | 12 | 52.17 |
| Other specified congenital malformations of the heart | Q24.8 | 31 | 0.004(0.002–0.005) | 17 | 54.84 |
| Congenital malformation of the heart, unspecified | Q24.9 | 31 | 0.004(0.002–0.005) | 14 | 45.16 |
| Other | Other codes of Q24 | 25 | 0.003(0.002–0.004) | 13 | 52.00 |
| **Congenital malformations of great arteries** | **Q25** | **1772** | **0.209(0.199–0.219)** | **170** | **9.59** |
| Patent ductus arteriosus | Q25.0 | 1550 | 0.183(0.174–0.192) | 15 | 0.97 |
| Coarctation of aorta | Q25.1 | 53 | 0.006(0.005–0.008) | 33 | 62.26 |
| Stenosis of aorta | Q25.3 | 63 | 0.007(0.006–0.009) | 46 | 73.02 |
| Stenosis of the pulmonary artery | Q25.6 | 77 | 0.009(0.007–0.011) | 49 | 63.64 |
| Other | Other codes of Q25 | 92 | 0.011(0.009–0.013) | 40 | 43.48 |
| **Congenital malformations of great veins** | **Q26** | **237** | **0.028(0.024–0.032)** | **104** | **43.88** |
| Persistent left superior vena cava | Q26.1 | 197 | 0.023(0.020–0.026) | 87 | 44.16 |
| Partial anomalous pulmonary venous connection | Q26.3 | 12 | 0.001(0.001–0.002) | 5 | 41.67 |
| Anomalous portal venous connection | Q26.5 | 21 | 0.002(0.001–0.004) | 5 | 23.81 |

(*Continued*)

**Table 2.** (Continued)

| Types | ICD codes | n | Prevalence (%, 95% CI) | Perinatal deaths (n) | PMR of specific CHDs (%) * |
|---|---|---|---|---|---|
| Other | Other codes of Q26 | 11 | 0.001(0.001–0.002) | 7 | 63.64 |

**Abbreviation**: PMR = perinatal mortality rate; CHD = congenital heart defect; ICD = International classification of diseases; CI = confidence intervals

* **Note**: 52.15% (2170 / 4161) of cases had multiple specific CHDs

**Table 3. Results of the univariate analysis.**

| Variables | CHDs (n) | Perinatal deaths (n) | PMR (%, 95%CI) | Unadjusted OR (95%CI) |
|---|---|---|---|---|
| Sex | | | | |
| Male | 2204 | 514 | 23.32(21.31–25.34) | Reference |
| Female | 1952 | 457 | 23.41(21.27–25.56) | 1.01(0.87–1.16) |
| Unknown | 5 | 5 | - | - |
| Residence | | | | |
| Urban | 2342 | 328 | 14.01(12.49–15.52) | Reference |
| Rural | 1819 | 648 | 35.62(32.88–38.37) | 3.40(2.92–3.95) |
| Maternal age (years old) | | | | |
| 25–29 | 1700 | 388 | 22.82(20.55–25.09) | Reference |
| <20 | 69 | 32 | 46.38(30.31–62.45) | 2.92(1.80–4.76) |
| 20–24 | 474 | 176 | 37.13(31.65–42.62) | 2.00(1.61–2.48) |
| 30–34 | 1270 | 248 | 19.53(17.10–21.96) | 0.82(0.69–0.98) |
| > = 35 | 648 | 132 | 20.37(16.90–23.85) | 0.87(0.69–1.08) |
| Maternal education level | | | | |
| Senior high school | 1540 | 396 | 25.71(23.18–28.25) | Reference |
| Junior high school and below | 587 | 201 | 34.24(29.51–38.98) | 1.50(1.23–1.85) |
| University and above | 2034 | 379 | 18.63(16.76–20.51) | 0.66(0.56–0.78) |
| Number of births | | | | |
| Singletons | 3849 | 943 | 24.50(22.94–26.06) | Reference |
| Multiple births | 312 | 33 | 10.58(6.97–14.19) | 0.36(0.25–0.53) |
| Number of pregnancies (times) | | | | |
| 1 (first pregnancy) | 1299 | 262 | 20.17(17.73–22.61) | Reference |
| 2 | 1186 | 295 | 24.87(22.04–27.71) | 1.31(1.08–1.58) |
| 3 | 808 | 186 | 23.02(19.71–26.33) | 1.18(0.96–1.46) |
| 4 | 473 | 126 | 26.64(21.99–31.29) | 1.44(1.13–1.84) |
| > = 5 | 395 | 107 | 27.09(21.96–32.22) | 1.47(1.13–1.91) |
| Parity | | | | |
| 1 (first birth) | 2160 | 516 | 23.89(21.83–25.95) | Reference |
| 2 | 1756 | 397 | 22.61(20.38–24.83) | 0.93(0.80–1.08) |
| > = 3 | 245 | 63 | 25.71(19.36–32.06) | 1.10(0.81–1.49) |
| Time of diagnosis | | | | |
| Postnatal period (within 7 days) | 2702 | 15 | 0.56(0.27–0.84) | Reference |
| Prenatal period (> = 37 weeks) | 140 | 48 | 34.29(24.59–43.99) | 93.46(50.48–173.03) |
| Prenatal period (33–36 weeks) | 278 | 184 | 66.19(56.62–75.75) | 350.64(199.28–616.97) |
| Prenatal period (< = 32 weeks) | 1041 | 729 | 70.03(64.95–75.11) | 418.55(247.72–707.20) |

**Abbreviation**: PMR = perinatal mortality rate; CHD = congenital heart defect; CI = confidence intervals; OR = crude odds ratio

**Table 4. Results of the multivariable logistic regression analysis.**

| Variables | Regression coefficient | Standard deviation | Wald $\chi^2$ | P | Adjusted OR (95%CI) |
|---|---|---|---|---|---|
| Residence | | | | | |
| Urban | | | | | Reference |
| Rural | 0.79 | 0.12 | 45.81 | <0.001 | 2.21(1.76–2.78) |
| Maternal age (years old) | | | | | |
| 25–29 | | | | | Reference |
| <20 | 0.88 | 0.42 | 4.35 | 0.04 | 2.40(1.05–5.47) |
| 20–24 | 0.76 | 0.19 | 15.32 | <0.001 | 2.13(1.46–3.11) |
| 30–34 | -0.29 | 0.14 | 4.18 | 0.04 | 0.75(0.56–0.99) |
| > = 35 | -0.41 | 0.19 | 4.84 | 0.03 | 0.66(0.46–0.96) |
| Number of births | | | | | |
| Singletons | | | | | Reference |
| Multiple births | -0.74 | 0.26 | 8.01 | 0.005 | 0.48(0.28–0.80) |
| Number of pregnancies (times) | | | | | |
| 1 (first pregnancy) | | | | | Reference |
| 2 | 0.47 | 0.16 | 9.14 | 0.002 | 1.60(1.18–2.18) |
| 3 | 0.36 | 0.18 | 4.18 | 0.04 | 1.43(1.01–2.02) |
| 4 | 0.61 | 0.21 | 8.25 | 0.004 | 1.84(1.21–2.78) |
| > = 5 | 0.70 | 0.23 | 9.23 | 0.002 | 2.02(1.28–3.18) |
| Time of diagnosis | | | | | |
| Postnatal period (within 7 days) | | | | | Reference |
| Prenatal period (> = 37 weeks) | 4.35 | 0.32 | 185.44 | <0.001 | 77.37(41.37–144.67) |
| Prenatal period (33–36 weeks) | 5.72 | 0.29 | 385.38 | <0.001 | 305.63(172.61–541.15) |
| Prenatal period (< = 32 weeks) | 5.98 | 0.27 | 491.70 | <0.001 | 395.69(233.23–671.33) |
| Constant | -5.83 | 0.29 | 397.22 | <0.001 | |

**Abbreviations**: CHD = congenital heart defect; OR = odds ratio; CI = confidence intervals

most recent systematic study to report PMRs of a broad range of specific CHDs, and on the association between demographic characteristics and perinatal deaths attributable to CHDs in China using multivariable logistic regression analysis. Our findings may contribute to the field.

First, the PMR of CHDs in this study was relatively high (23.46%). The PMR of CHDs was 3.6% in Europe (2000–2005) [19] and 18.3% in Zhejiang Province (a developed province in eastern China) (2014–2018) [6]. The PMR of CHDs was significantly higher in this study than in Europe and Zhejiang Province. It may be directly associated with the medical TOP rate, as the medical TOP rate was 22.49% in this study, 17.04% in Zhejiang Province [6], and 5.6% in Europe [19]. Although there are relatively few studies reporting the PMR of CHDs, we believe that the PMR of CHDs may be mainly related to medical and economic conditions because advanced medical conditions are good for the diagnosis and treatment of CHDs. Better economic conditions are good for the survival of children with CHD [19, 23, 24]. Hunan Province is a relatively undeveloped province in China, and the relatively high PMR of CHDs may be associated with relatively undeveloped economic and medical conditions [25]. In addition, the PMR of CHDs showed a downward trend from 2016 to 2020. It may be mainly related to developing medical and economic conditions, as shown in the Hunan Statistical Yearbook [25].

In this study, we systematically reported the PMRs of a broad range of specific CHDs. To our knowledge, relatively few studies have reported the PMR of specific CHDs [26]. In

addition, we also obtained several interesting findings from the PMRs of specific CHDs. For example, although many specific CHDs are not usually considered severe, they have relatively high PMR, such as ventricular septal defect [27, 28]. It is associated with the fact that most specific CHDs combined with other defects [29], and fetuses with multiple birth defects may be more likely to end in death [30].

Second, in this study, the PMR of CHDs was higher in rural than urban areas. Few studies have reported the PMR of CHDs by residence. However, many studies have reported a higher prevalence of CHDs in urban areas than in rural areas [31–33]. We believe that the higher PMR of CHDs in rural than urban areas may be mainly related to better medical and economic conditions in urban areas than in rural areas, which may cause a higher TOP rate of CHDs in rural areas than in urban areas, similar to the above discussion. We believe that the higher prevalence of CHDs in urban areas than in rural areas may also partly result from the higher PMR of CHDs in rural areas than in urban areas. In addition, factors other than medical and economic conditions, such as health knowledge and behaviors, may also contribute to the higher PMR of CHDs in rural areas than in urban areas. For example, Qun et al. found that the proportions of obesity, smoking or social drug use or alcohol consumption during pregnancy, and mental health illness in pre-pregnancy or during pregnancy were significantly higher in rural than in urban areas [34], which may contribute to perinatal deaths attributable to CHDs [35, 36]. In addition, by comparing the results from the univariate and multivariable logistic regression analysis, we found that mutual adjustment did not affect the associations considerably, apart from the observation that the OR for rural areas was somewhat attenuated. It suggests that this analysis apparently covered some, but not all aspects which account for higher PMRs in rural areas.

Third, perinatal deaths attributable to CHDs were associated with low maternal age, which may be primarily associated with the higher prevalence of severe CHDs in the low maternal age group. For example, Mamasoula et al. found an increased total prevalence of very severe CHDs (including double outlet right ventricle, hypoplastic left heart syndrome, hypoplastic right heart syndrome, atrioventricular septal defect, coarctation of aorta and atrial septal defect) in younger mothers compared to those aged 25–29 years [37].

Fourth, perinatal deaths attributable to CHDs were associated with a high number of pregnancies. It may also be mainly associated with the higher prevalence of severe CHDs among pregnant women with a high number of pregnancies. Previous studies have shown that a high number of pregnancies may be mainly related to spontaneous miscarriage (including many recurrent miscarriages), which may be associated with many severe birth defects [38]. In addition, miscarriage may indicate that the pregnant woman has many other disorders, such as parental chromosomal anomalies, maternal thrombophilic disorders, immune dysfunction, and various endocrine disturbances [39], which may also contribute to perinatal deaths.

Fifth, perinatal deaths attributable to CHDs were more common in CHDs diagnosed prenatally (especially at low gestational age), which may be mainly associated with consideration of maternal health. Generally, suppose a pregnant woman has a fetus with severe birth defects (which is not survivable). In that case, they choose to terminate the pregnancy as soon as possible to minimize the adverse health effects [40, 41]. In addition, with the development of prenatal screening and diagnosis of CHDs, an increasing number of CHDs are diagnosed and terminated before 28 weeks of gestational age, which is not included in perinatal deaths [6, 42].

Sixth, perinatal deaths attributable to CHDs were less common in multiple births than in singletons. It is inconsistent with common sense. There are a few similar reports. For example, previous studies have shown that most multiple births were associated with assisted reproductive technology [43], and adverse pregnancy outcomes, including birth defects and perinatal deaths, were more common in multiple births [44–46]. However, in this study, perinatal

deaths attributable to CHDs were less common in multiple births than in singletons. It may be mainly associated with parental attitudes towards CHDs. Most parents who conceive by assisted reproductive technology may have reproductive difficulties, and they are more willing to give birth to fetuses than those who conceive naturally, even if the fetuses are born with birth defects [47], which may explain the lower PMR of CHDs in multiple births than in singletons.

Some things could be improved. First, most cases had multiple specific CHDs, and many CHDs were combined with other defects. However, we did not analyze these multiple defects in depth due to data and analytic method limitations. Second, some demographic characteristics that may be associated with perinatal deaths attributable to CHDs were not included in this study due to data limitations, such as paternal age, maternal body mass index, gestational weight gain, and gestational diabetes. Third, our findings showed an association, but not a causal relationship, between perinatal deaths attributable to CHDs and a broad range of demographic characteristics, which need to be further investigated in the future. Fourth, CHDs before 28 weeks of gestation or within 7 days of birth were not included in this study due to data limitations.

## Conclusion

Perinatal deaths were common in CHDs in Hunan in 2016–2020. Several demographic characteristics were associated with perinatal deaths attributable to CHDs, which may be summarized mainly as economic and medical conditions, severity of CHDs, and parental attitudes toward CHDs.

## Acknowledgments

The authors thank the staff working for the Birth Defects Surveillance System of Hunan Province, China, from 2016 to 2020.

## Author Contributions

**Conceptualization:** Xu Zhou, Haiyan Kuang.

**Data curation:** Xu Zhou, Jian He.

**Formal analysis:** Xu Zhou, Haiyan Kuang.

**Methodology:** Haiyan Kuang.

**Project administration:** Xu Zhou, Haiyan Kuang, Hua Wang.

**Supervision:** Junqun Fang, Hua Wang.

**Validation:** Xu Zhou.

**Visualization:** Xu Zhou.

**Writing – original draft:** Xu Zhou, Haiyan Kuang.

**Writing – review & editing:** Xu Zhou, Haiyan Kuang, Junqun Fang, Hua Wang.

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
