## [Decision Letter · Decision Letter 0]

17 Jan 2024

PONE-D-23-39650Perinatal deaths from congenital heart defects in Hunan Province, China, 2016-2020PLOS ONE

Dear Dr. Zhou,

Thank you for submitting your manuscript to PLOS ONE. After careful consideration, we feel that it has merit but does not fully meet PLOS ONE’s publication criteria as it currently stands. Therefore, we invite you to submit a revised version of the manuscript that addresses the points raised during the review process.

There are two major issues which need to be addressed: 

- It is unclear which conclusions can be drawn from this analysis, as the perinatal mortality in CHD children seems mainly driven by selective TOPs. It might therefore be more appropriate to re-do the whole analysis with selective TOP instead of perinatal mortality as the main outcome, and to discuss the strikingly high rates of selective TOP in CHD pregnancies in Hunan (or China). Results for perinatal mortality might be shown within a sensitivity analysis.

 - Much more information is needed to understand the design of the Birth Defects Surveillance System, e.g. how, when and where where the participating women exactly recruited, what was the participation rate, did the participating and non-participating women differ with respect to demographic characteristics and birth outcomes, how were they followed up with respect to their birth outcomes, was there loss of follow-up, which information was collected and how etc...

Please find further (minor) points below.

We look forward to receiving your revised manuscript.

Kind regards,

Andreas Beyerlein

Academic Editor

PLOS ONE

Journal Requirements:

https://bmcpregnancychildbirth.biomedcentral.com/articles/10.1186/s12884-023-06092-5

https://www.nature.com/articles/s41598-023-47741-1

In your revision ensure you cite all your sources (including your own works), and quote or rephrase any duplicated text outside the methods section. Further consideration is dependent on these concerns being addressed.

Additional Editor Comments:

Whole manuscript:

- Please replace "multivariate" by "multivariable".

- Speaking of "risk factors" and "protective factors" is somewhat misleading, as the authors show associations, but not causal relations.

- The interpretation of the ORs is not straightforward and should be explained shortly both in the Abstract and in the main text.

- The first and last sentence of the Conclusions (both in the abstract and twice in the main text) are superfluous and should therefore be omitted.

- Suggest to replace per thousand by percent throughout the manuscript (including table 1) to make the proportion numbers directly comparable to each other.

- In the spirit of Open and Reproducible Science, the analysis code should be made available in an online repository together with a data dictionary, and the respective URL should be mentioned in the Methods section.

Abstract:

- The sentence "Multivariate logistic regression analysis showed..." is awkward and should be revised, e.g. "In stepwise logistic regression analysis, the following variables were selected for the final model: ..."

- ORs and 95% CIs should be reported for all predictors; p-values should be rather left out.

- It is difficult to understand the meaning of "diagnosed in the postnatal period" and "low gestational age of diagnosis". This should be described in a clearer way.

Main text:

- l. 60: Suggest to replace "were" by "are".

- l. 63: What is meant by "accepted prevalence"?

- l. 70: Abbreviation "WHO" should be explained at first appearance.

- l. 74-93: Instead of just counting up these studies, they should be set in context to each other and to the research question. The same applies for the limitations.

- l. 106-109: What was the rationale to choose exactly these 8 variables? Ideally, some references should be added here. Additionally, this sentence should rather be put into the "statistical analysis" section.

- Maternal BMI, gestational weight gain as well as gestational and pre-gestational diabetes should be added as potential predictors, if available. If not, the lack of these variables should be discussed as a potential limitation.

- l. 120-122: How (and why) were the anonymized data deidentified?

- l. 122: Which experiments are meant here?

- Please add a table showing demographic characteristics (including gestational age and numbers of preterm births) for the whole dataset. Were there any missing data in these variables, or if not, why not?

- Table 1: It would be helpful to the reader to explain the difference of "PMR of" and "PMR attributable to" (or TOP instead of PMR in the revised version) in the table legend.

- Table 4: This information should be given in the main text instead of adding a table for this.

- l. 289-294: This paragraph is weak and should be replaced by a thorough discussion about limitations and strengths of this analysis.

- Additionally, another paragraph might be added which discusses a) potential ways to further reduce PMR / TOP in CHD pregnancies in China and b) the generalizability of these results to whole China and to other countries.

Reviewers' comments:

Reviewer's Responses to Questions

**Comments to the Author**

1. Is the manuscript technically sound, and do the data support the conclusions?

Reviewer #1: Partly

Reviewer #2: Yes

2. Has the statistical analysis been performed appropriately and rigorously? 

Reviewer #1: Yes

Reviewer #2: Yes

3. Have the authors made all data underlying the findings in their manuscript fully available?

Reviewer #1: Yes

Reviewer #2: Yes

4. Is the manuscript presented in an intelligible fashion and written in standard English?

Reviewer #1: Yes

Reviewer #2: Yes

5. Review Comments to the Author

Reviewer #1: The manuscript described the perinatal mortality rates (PMR) of congenital heart defects (CHDs) and identified risk factors for perinatal deaths attributable to CHDs. The prevalence of CHDs was 4.91‰ (95%CI: 4.76-5.06).The total PMR was 0.88% (95%CI: 0.86-0.90), and the PMR of CHDs was 23.46%. This manuscript provides an appropriate study design and performance.

Major points

1.I suggest the author to provide more specific statistical methods and results，e.g. the inclusion criteria of variables in logistic regression analysis, VIF between variables? p values in tables?

2. Cases of BD included CHD?

3. Is there difference between years?

Reviewer #2: This is a work on the perinatal deaths from congenital heart defects in Hunan Province, China. The authors describe the perinatal mortality rates of congenital heart defects and use multivariate logistic regression to identify possible risk factors.

The manuscript and the results are well organised. Please find my comments below:

- Lines 41-49: Mulitvariate analysis results should be more consistent. Please use ORs, 95%Cis and p values (in their exact values) to describe the multivariate model estimates.

- Lines 74-93: The first name of the author is used when mentioned the references 16, 17, 14. Please revise.

- Line 146: The reference for SPSS version 18.0 is wrong. For versions 18 and earlier "SPSS Inc. in Chicago" should be used instead of "IBM Corp., NY, USA". (can be found in: https://www.ibm.com/support/pages/how-cite-ibm-spss-statistics-or-earlier-versions-spss).

- Lines 189-190 (and throught the text): Please use the exact value of p value (only if p=0.000 then it should be given as p<0.001).

- Lines 197-206 Same as the first comment, please use ORs, 95%Cis and p values (in their exact values) to describe the multivariate model estimates.

- Table 5: P values should be given in their exact values with three decimals. Again, if a p values=0.00 then it should be given as p<0.001.

6. PLOS authors have the option to publish the peer review history of their article (what does this mean?). If published, this will include your full peer review and any attached files.

Reviewer #1: No

Reviewer #2: No

---

## [Author Response · Author response to Decision Letter 0]

27 Feb 2024

PONE-D-23-39650

Perinatal deaths from congenital heart defects in Hunan Province, China, 2016-2020

PLOS ONE

There are two major issues which need to be addressed: 

- It is unclear which conclusions can be drawn from this analysis, as the perinatal mortality in CHD children seems mainly driven by selective TOPs. It might therefore be more appropriate to re-do the whole analysis with selective TOP instead of perinatal mortality as the main outcome, and to discuss the strikingly high rates of selective TOP in CHD pregnancies in Hunan (or China). Results for perinatal mortality might be shown within a sensitivity analysis.

Response: Thanks for your suggestion very much. 

According to your suggestion, we have done some revisions, but we did not re-do the whole analysis with selective TOP instead of perinatal mortality as the main outcome. The following are our main reasons: 

1. Although the perinatal mortality in CHD children seems mainly driven by selective TOPs, the main cause of perinatal deaths was CHDs or other severe birth defects, not selective TOP. These perinatal deaths would not have been preventable without the selective TOP. And the primary reason doctors adopt selective TOP is to alleviate the suffering of the pregnant woman and their families.

2. Selective TOP is also a sensitive topic in China, so we have tried our best to avoid statements about it in the manuscript.

If further revisions are required, we will do so as soon as possible.

- Much more information is needed to understand the design of the Birth Defects Surveillance System, e.g. how, when and where where the participating women exactly recruited, what was the participation rate, did the participating and non-participating women differ with respect to demographic characteristics and birth outcomes, how were they followed up with respect to their birth outcomes, was there loss of follow-up, which information was collected and how etc...

Response: Thanks for your suggestion very much. 

According to your suggestion, we have revised the manuscript in lines 92-151 (Methods)

https://bmcpregnancychildbirth.biomedcentral.com/articles/10.1186/s12884-023-06092-5

https://www.nature.com/articles/s41598-023-47741-1

In your revision ensure you cite all your sources (including your own works), and quote or rephrase any duplicated text outside the methods section. Further consideration is dependent on these concerns being addressed.

Response: Thanks for your suggestion very much. 

According to your suggestion, we have revised the manuscript in lines 52-91 (Introduction), 152-224 (Results), and 231-320 (Discussion).

Response: Thanks for your suggestion very much. 

According to your suggestion, we have revised the manuscript in lines 119-123 (Ethics approval and consent to participate) and 336-338 (Data Availability).

Additional Editor Comments:

Whole manuscript:

- Please replace "multivariate" by "multivariable".

Response: Thanks for your suggestion very much. 

According to your suggestion, we have revised the manuscript in lines 26, 33, 82, 86, 143, 203, 223, 226.

- Speaking of "risk factors" and "protective factors" is somewhat misleading, as the authors show associations, but not causal relations.

Response: Thanks for your suggestion very much. 

According to your suggestion, we have revised the manuscript in lines 27-28, 75-76, and 143-147.

- The interpretation of the ORs is not straightforward and should be explained shortly both in the Abstract and in the main text.

Response: Thanks for your suggestion very much. 

According to your suggestion, we have revised the manuscript in lines 20-28 (Abstract - Methods) and 139-151 (Abstract - Statistical analysis).

- The first and last sentence of the Conclusions (both in the abstract and twice in the main text) are superfluous and should therefore be omitted.

Response: Thanks for your suggestion very much. 

According to your suggestion, we have revised the manuscript in lines 321-327 (Conclusion).

- Suggest to replace per thousand by percent throughout the manuscript (including table 1) to make the proportion numbers directly comparable to each other.

Response: Thanks for your suggestion very much. 

According to your suggestion, we have revised the manuscript in lines 166-167 (Table 1) and 182-185 (Table 2).

- In the spirit of Open and Reproducible Science, the analysis code should be made available in an online repository together with a data dictionary, and the respective URL should be mentioned in the Methods section.

Response: Thanks for your suggestion very much. 

According to your suggestion, we have revised the manuscript in lines 139-151 (Statistical analysis) and 336-338 (Data Availability).

Abstract:

- The sentence "Multivariate logistic regression analysis showed..." is awkward and should be revised, e.g. "In stepwise logistic regression analysis, the following variables were selected for the final model: ..."

Response: Thanks for your suggestion very much. 

According to your suggestion, we have revised the manuscript in lines 33-35 (Abstract) and 213-215 (Results).

- ORs and 95% CIs should be reported for all predictors; p-values should be rather left out.

Response: Thanks for your suggestion very much. 

According to your suggestion, we have revised the manuscript in lines 29-44 (Abstract - Results) and 186-230 (Results).

- It is difficult to understand the meaning of "diagnosed in the postnatal period" and "low gestational age of diagnosis". This should be described in a clearer way.

Response: Thanks for your suggestion very much. 

According to your suggestion, we have revised the manuscript in lines 234 and 293. 

Main text:

- l. 60: Suggest to replace "were" by "are".

Response: Thanks for your suggestion very much. 

According to your suggestion, we have revised the manuscript in lines 53-54.

- l. 63: What is meant by "accepted prevalence"?

Response: Thanks for your suggestion very much. 

According to your suggestion, we have revised the manuscript in lines 55-56.

- l. 70: Abbreviation "WHO" should be explained at first appearance.

Response: Thanks for your suggestion very much. 

According to your suggestion, we have revised the manuscript in lines 62-64.

- l. 74-93: Instead of just counting up these studies, they should be set in context to each other and to the research question. The same applies for the limitations.

Response: Thanks for your suggestion very much. 

According to your suggestion, we have revised the manuscript in lines 66-87.

- l. 106-109: What was the rationale to choose exactly these 8 variables? Ideally, some references should be added here. Additionally, this sentence should rather be put into the "statistical analysis" section.

Response: Thanks for your suggestion very much. 

According to your suggestion, we have revised the manuscript in lines 147-150 (Methods - Statistical analysis) and 206-213 (Results).

- Maternal BMI, gestational weight gain as well as gestational and pre-gestational diabetes should be added as potential predictors, if available. If not, the lack of these variables should be discussed as a potential limitation.

Response: Thanks for your suggestion very much. 

According to your suggestion, we have revised the manuscript in lines 310-320 (Discussion - potential limitation).

- l. 120-122: How (and why) were the anonymized data deidentified?

Response: Thanks for your suggestion very much. 

According to your suggestion, we have revised the manuscript in lines 118-132 (Ethics approval and consent to participate).

- l. 122: Which experiments are meant here?

Response: Thanks for your suggestion very much. 

According to your suggestion, we have revised the manuscript in lines 118-132 (Ethics approval and consent to participate).

- Please add a table showing demographic characteristics (including gestational age and numbers of preterm births) for the whole dataset. Were there any missing data in these variables, or if not, why not?

Response: Thanks for your suggestion very much. 

According to your suggestion, we have revised the manuscript in lines 202-204 (Table 3. Results of the univariate analysis). 

Some demographic characteristics were not in the table due to data limitation, such as variables were not included in the case cards or missing data.

- Table 1: It would be helpful to the reader to explain the difference of "PMR of" and "PMR attributable to" (or TOP instead of PMR in the revised version) in the table legend.

Response: Thanks for your suggestion very much. 

According to your suggestion, we have revised the manuscript in lines 71, 72, 81, 153, 166, 168, 181, 182, 187, 193, 195, 198, 239-243, 252, 253, 264, 268, 279, and 325. 

In the revised manuscript, we revised all "PMR attributable to" to "PMR of".

- Table 4: This information should be given in the main text instead of adding a table for this.

Response: Thanks for your suggestion very much. 

As it is rather cumbersome to express it in words, we propose to keep the table.

If further revisions are required, we will do so as soon as possible.

- l. 289-294: This paragraph is weak and should be replaced by a thorough discussion about limitations and strengths of this analysis.

Response: Thanks for your suggestion very much. 

According to your suggestion, we have revised the manuscript in lines 310-320 (limitations) and 232-238 (strengths).

And in the Introduction and Discussion sections, we also presented the strengths of this study.

- Additionally, another paragraph might be added which discusses a) potential ways to further reduce PMR / TOP in CHD pregnancies in China and b) the generalizability of these results to whole China and to other countries.

Response: Thanks for your suggestion very much. 

According to your suggestion, we have revised the manuscript in lines 324-327.

Reviewers' comments:

Reviewer's Responses to Questions

Comments to the Author

5. Review Comments to the Author

Reviewer #1: The manuscript described the perinatal mortality rates (PMR) of congenital heart defects (CHDs) and identified risk factors for perinatal deaths attributable to CHDs. The prevalence of CHDs was 4.91‰ (95%CI: 4.76-5.06).The total PMR was 0.88% (95%CI: 0.86-0.90), and the PMR of CHDs was 23.46%. This manuscript provides an appropriate study design and performance.

Major points

1.I suggest the author to provide more specific statistical methods and results，e.g. the inclusion criteria of variables in logistic regression analysis, VIF between variables? p values in tables?

Response: Thanks for your suggestion very much. 

According to your suggestion, we have revised the manuscript in lines 139-151 (Statistical analysis) and 152-230 (Results).

2. Cases of BD included CHD?

Response: Thanks for your suggestion very much. 

According to your suggestion, we have revised the manuscript in lines 153-165.

In this study, cases of BD included CHD.

3. Is there difference between years?

Response: Thanks for your suggestion very much. 

According to your suggestion, we have revised the manuscript in lines 163-165 (Results).

In this study, there difference in PMRs of CHDs between years.

Reviewer #2: This is a work on the perinatal deaths from congenital heart defects in Hunan Province, China. The authors describe the perinatal mortality rates of congenital heart defects and use multivariate logistic regression to identify possible risk factors.

The manuscript and the results are well organised. Please find my comments below:

- Lines 41-49: Mulitvariate analysis results should be more consistent. Please use ORs, 95%Cis and p values (in their exact values) to describe the multivariate model estimates.

Response: Thanks for your suggestion very much. 

According to your suggestion, we have revised the manuscript in lines 29-44 (Abstract - Results).

- Lines 74-93: The first name of the author is used when mentioned the references 16, 17, 14. Please revise.

Response: Thanks for your suggestion very much. 

According to your suggestion, we have revised the manuscript in lines 69-78.

- Line 146: The reference for SPSS version 18.0 is wrong. For versions 18 and earlier "SPSS Inc. in Chicago" should be used instead of "IBM Corp., NY, USA". (can be found in: https://www.ibm.com/support/pages/how-cite-ibm-spss-statistics-or-earlier-versions-spss).

Response: Thanks for your suggestion very much. 

According to your suggestion, we have revised the manuscript in lines 151.

- Lines 189-190 (and throught the text): Please use the exact value of p value (only if p=0.000 then it should be given as p<0.001).

Response: Thanks for your suggestion very much. 

According to your suggestion, we have revised the manuscript in lines 152-230 (Results).

- Lines 197-206 Same as the first comment, please use ORs, 95%Cis and p values (in their exact values) to describe the multivariate model estimates.

Response: Thanks for your suggestion very much. 

According to your suggestion, we have revised the manuscript in lines 206-234.

- Table 5: P values should be given in their exact values with three decimals. Again, if a p values=0.00 then it should be given as p<0.001.

Response: Thanks for your suggestion very much. 

According to your suggestion, we have revised the manuscript in lines 228-230 (Table 5).

---

## [Decision Letter · Decision Letter 1]

21 Mar 2024

PONE-D-23-39650R1Perinatal deaths attributable to congenital heart defects in Hunan Province, China, 2016-2020PLOS ONE

Dear Dr. Zhou,

Thank you for submitting your manuscript to PLOS ONE. After careful consideration, we feel that it has merit but does not fully meet PLOS ONE’s publication criteria as it currently stands. Therefore, we invite you to submit a revised version of the manuscript that addresses the points raised during the review process.

We look forward to receiving your revised manuscript.

Kind regards,

Andreas Beyerlein

Academic Editor

PLOS ONE

Journal Requirements:

Additional Editor Comments:

The authors have done well in revising and improving their manuscript. However, a few issues remain to be addressed, please see below:

- Abstract: Suggest to write "In stepwise logistic regression analysis, perinatal deaths...", i.e. omit the statement which variables were selected.

- l. 38-44: Rather than giving the ORs of all categories it might be more relevant to the reader to mention the associations of PMR with other variables such as time of diagnosis.

- Unfortunately, the paragraph in l. 66-87 is still weak both with respect to English wording and with respect to content. It is also unclear what is meant by "unrepresentative hospitals" or "some studies need to be updated". The text might improve if the authors will remove the counting ("first" to "fifth") and integrate the references mentioned before directly into their arguments.

- In general, proof-reading of the whole manuscript by an English native speaker might be advisable.

- l. 96: What were the criterions to define "representative hospitals"? Out of how many hospitals were these 52 hospitals chosen?

- l. 189-192: Suggest to write "Compared to senior high school as maternal education level..."

- l. 38-40 and 193: add "years"

- l. 193-194: These PMRs are not attributed to the age categories in the same way as in table 3.

- In general, the text should repeat only a small selection of the numbers given in the tables. It is enough to mention the associations in a qualitative way (e.g. "maternal age was negatively associated with PMR...") and refer to the respective tables. All numbers which remain in the text should be thoroughly checked for consistence with the tables.

- l. 196: The weeks should be put out of the brackets, as the sentence makes no sense without this information.

- Table 4 is superfluous.

- l. 216-224: It would be sufficient to mention that the results from table 5 were quite similar to those from table 3, i.e. mutual adjustment did not affect the associations considerably, apart from the obervsation that the OR for rural area was somewhat attenuated. The latter might be mentioned in the discussion with the interpretation that this analysis apparently covered some, but not all aspects which account for higher PMRs in rural areas.

- l. 235: Suggest to add "To our knowledge,..."

- l. 239-241: This result needs to be set in context. Is Hunan a less developed area in China?

- l. 248-250: Is there any evidence to support this statement, e.g. other medical improvements during this period in Hunan?

- l. 265-266 and 293-295: I don't understand these sentences, please explain in more detail.

- l. 279-283: This argument seems to contradict your argument from the Response Letter that these perinatal deaths would not have been preventable without the selective TOP. Please explain.

- l. 64-65, 207-213, 239-241, 257-263, 298, 324-327: These sentences seem superfluous.

- The Methods section still lacks an URL were the analysis code can be found.

Reviewers' comments:

Reviewer's Responses to Questions

**Comments to the Author**

1. If the authors have adequately addressed your comments raised in a previous round of review and you feel that this manuscript is now acceptable for publication, you may indicate that here to bypass the “Comments to the Author” section, enter your conflict of interest statement in the “Confidential to Editor” section, and submit your "Accept" recommendation.

Reviewer #1: All comments have been addressed

Reviewer #2: All comments have been addressed

2. Is the manuscript technically sound, and do the data support the conclusions?

Reviewer #1: Yes

Reviewer #2: Yes

3. Has the statistical analysis been performed appropriately and rigorously? 

Reviewer #1: Yes

Reviewer #2: Yes

4. Have the authors made all data underlying the findings in their manuscript fully available?

Reviewer #1: Yes

Reviewer #2: Yes

5. Is the manuscript presented in an intelligible fashion and written in standard English?

Reviewer #1: Yes

Reviewer #2: Yes

6. Review Comments to the Author

Reviewer #1: In my opinion, in the revision process of their manuscript, the authors have adressed the points I raised.

I feel there might be some flaws in the language used in the new passages.

Reviewer #2: All my comments have been adequately addressed in the author’s response, I have no further comments.

7. PLOS authors have the option to publish the peer review history of their article (what does this mean?). If published, this will include your full peer review and any attached files.

Reviewer #1: **Yes: **xiaoying zhang

Reviewer #2: No

---

## [Author Response · Author response to Decision Letter 1]

18 Apr 2024

PONE-D-23-39650R1

Perinatal deaths attributable to congenital heart defects in Hunan Province, China, 2016-2020

Additional Editor Comments:

The authors have done well in revising and improving their manuscript. However, a few issues remain to be addressed, please see below:

- Abstract: Suggest to write "In stepwise logistic regression analysis, perinatal deaths...", i.e. omit the statement which variables were selected.

Response: Thanks for your suggestion very much. 

According to your suggestion, we revised it in lines 37-38.

- l. 38-44: Rather than giving the ORs of all categories it might be more relevant to the reader to mention the associations of PMR with other variables such as time of diagnosis.

Response: Thanks for your suggestion very much. 

According to your suggestion, we revised it in lines 38-42.

- Unfortunately, the paragraph in l. 66-87 is still weak both with respect to English wording and with respect to content. It is also unclear what is meant by "unrepresentative hospitals" or "some studies need to be updated". The text might improve if the authors will remove the counting ("first" to "fifth") and integrate the references mentioned before directly into their arguments.

Response: Thanks for your suggestion very much. 

According to your suggestion, we revised it in lines 63-83.

- In general, proof-reading of the whole manuscript by an English native speaker might be advisable.

Response: Thanks for your suggestion very much. 

Due to time and financial constraints, we asked people who have lived and studied in the United States to revise the entire text.

We hope that these revisions meet the requirements. 

If further revisions are required, we will do it as soon as possible.

- l. 96: What were the criterions to define "representative hospitals"? Out of how many hospitals were these 52 hospitals chosen?

Response: Thanks for your suggestion very much. 

According to your suggestion, we revised it in lines 90-98.

- l. 189-192: Suggest to write "Compared to senior high school as maternal education level..."

Response: Thanks for your suggestion very much. 

According to your suggestion, we revised it in lines 179-182.

- l. 38-40 and 193: add "years"

Response: Thanks for your suggestion very much. 

According to your suggestion, we revised it in lines 39-40 and 198-201.

- l. 193-194: These PMRs are not attributed to the age categories in the same way as in table 3.

Response: Thanks for your suggestion very much. 

According to your suggestion, we revised it in lines 183-185.

- In general, the text should repeat only a small selection of the numbers given in the tables. It is enough to mention the associations in a qualitative way (e.g. "maternal age was negatively associated with PMR...") and refer to the respective tables. All numbers which remain in the text should be thoroughly checked for consistence with the tables.

Response: Thanks for your suggestion very much. 

According to your suggestion, we revised it in lines 183-185.

- l. 196: The weeks should be put out of the brackets, as the sentence makes no sense without this information.

Response: Thanks for your suggestion very much. 

According to your suggestion, we revised it in lines 183-185.

- Table 4 is superfluous.

Response: Thanks for your suggestion very much. 

According to your suggestion, we deleted table 4.

- l. 216-224: It would be sufficient to mention that the results from table 5 were quite similar to those from table 3, i.e. mutual adjustment did not affect the associations considerably, apart from the obervsation that the OR for rural area was somewhat attenuated. The latter might be mentioned in the discussion with the interpretation that this analysis apparently covered some, but not all aspects which account for higher PMRs in rural areas.

Response: Thanks for your suggestion very much. 

According to your suggestion, we revised it in lines 250-254.

- l. 235: Suggest to add "To our knowledge,..."

Response: Thanks for your suggestion very much. 

According to your suggestion, we revised it in lines 212-213.

- l. 239-241: This result needs to be set in context. Is Hunan a less developed area in China?

Response: Thanks for your suggestion very much. 

According to your suggestion, we revised it in lines 218-219 and 224-226.

- l. 248-250: Is there any evidence to support this statement, e.g. other medical improvements during this period in Hunan?

Response: Thanks for your suggestion very much. 

According to your suggestion, we revised it in lines 227-229.

- l. 265-266 and 293-295: I don't understand these sentences, please explain in more detail.

Response: Thanks for your suggestion very much. 

According to your suggestion, we revised it in lines 237-254 and 271-278.

- l. 279-283: This argument seems to contradict your argument from the Response Letter that these perinatal deaths would not have been preventable without the selective TOP. Please explain.

Response: Thanks for your suggestion very much. 

According to your suggestion, we revised it in lines 255-261.

- l. 64-65, 207-213, 239-241, 257-263, 298, 324-327: These sentences seem superfluous.

Response: Thanks for your suggestion very much. 

According to your suggestion, we deleted it. 

- The Methods section still lacks an URL were the analysis code can be found.

Response: Thanks for your suggestion very much. 

Multivariate logistic regression analysis of risk factors for birth defects: a study from population-based surveillance data - PubMed (nih.gov)

Epidemiology of congenital polydactyly and syndactyly in Hunan Province, China - PubMed (nih.gov)

We have published several similar articles and all the information needed has been provided.

Please give us more detailed tips.

---

## [Editor Report · Decision Letter 2]

26 Apr 2024

PONE-D-23-39650R2Perinatal deaths attributable to congenital heart defects in Hunan Province, China, 2016-2020PLOS ONE

Dear Dr. Zhou,

Thank you for submitting your manuscript to PLOS ONE. After careful consideration, we feel that it has merit but does not fully meet PLOS ONE’s publication criteria as it currently stands. Therefore, we invite you to submit a revised version of the manuscript that addresses the points raised during the review process.

**A number of issues still has to be addressed, please see a detailed list below.**

We look forward to receiving your revised manuscript.

Kind regards,

Andreas Beyerlein

Academic Editor

PLOS ONE

Additional Editor Comments:

- l. 23: The sentence should read "...whose mothers delivered...".

- l. 24-30: Suggest to condense these sentences as follows: "Surveillance data included demographic characteristics such as sex, residence, maternal age, and other key information, and were used to calculate prevalence of CHDs and perinatal mortality rates (PMR) with 95% confidence intervals (CI)."

- l. 40-41: "OR>1" is too imprecise and should be replaced by the calculated ORs with 95% CIs.

- l. 44 and 301: Suggest to write "Perinatal deaths were common in CHDs in Hunan in 2016-2020."

- l. 68: It should be made clear that reference 9 was written by the same authors based on the same dataset, and it should be explained in which aspects this new paper overlaps with the old one and what it adds.

- l. 83 (and other places): "Multivariate" should be replaced by "multivariable".

- l. 121-122: I still don't understand why and how patient records were "deidentified" given that they were anonymized anyway. Can this sentence be omitted? Otherwise pleas add some details.

- l. 131: Suggest to write "Prevalence and PMR of specific CHDs...".

- l. 149: Which Government? Suggest to write: "The Hunan Provincial Health Commission routinely collected surveillance data based on the Hunan Maternal and Child Health Surveillance Manual developed by...".

- l. 190-195: Suggest to omit the first two sentences and just write: "Starting with all variables from table 3, the following variables were selected for the final model in stepwise logistic regression analysis: ..."

- l. 196: In order to make the difference to the previous analysis clearer, the sentence might rather begin with "In the logistic regression model with mutual adjustment, perinatal deaths..."

- l. 217: It is strange to call this a meaningful finding given that more or less the same value has been mentioned as result of reference 9 (see also my comment above). The authors should be very clear in distinguishing new and old findings on these data in order to avoid self-plagiarism.

- It appears dubious that the results and discussion on selective TOPs have been deleted. I understand that selective TOPs are a sensitive topic in China, but the findings cannot be seriously interpreted without explaning this issue properly. In particular, the paragraph in line 217-229 speculates about potential reasons why the PMR of CHDs is so much higher in Hunan (and Zheijang) than in Europe, but these rates just seem to match the rates of selective TOPs. It would e.g. be relevant to mention the PMR for CHDs without selective TOPs and to discuss whether this is comparable to other countries. Further, it should be discussed why there were so much more selective TOPs in Hunan than in Europe, and whether these are more common in rural than urban areas (and if so, why).

- I was asking for setting up an online repository which covers both the analysis code and an accompanying data dictionary, see e.g. here: https://osf.io/qg4u6/

The respective URL should be mentioned in the Methods section.

---

## [Author Response · Author response to Decision Letter 2]

13 May 2024

PONE-D-23-39650R2

Perinatal deaths attributable to congenital heart defects in Hunan Province, China, 2016-2020

Additional Editor Comments:

- l. 23: The sentence should read "...whose mothers delivered...".

Response: Thanks very much for your suggestion. 

According to your suggestion, we revised it in lines 23-24.

- l. 24-30: Suggest to condense these sentences as follows: "Surveillance data included demographic characteristics such as sex, residence, maternal age, and other key information, and were used to calculate prevalence of CHDs and perinatal mortality rates (PMR) with 95% confidence intervals (CI)."

Response: Thanks very much for your suggestion. 

According to your suggestion, we revised it in lines 24-26.

- l. 40-41: "OR>1" is too imprecise and should be replaced by the calculated ORs with 95% CIs.

Response: Thanks very much for your suggestion. 

According to your suggestion, we revised it in lines 35-44.

- l. 44 and 301: Suggest to write "Perinatal deaths were common in CHDs in Hunan in 2016-2020."

Response: Thanks very much for your suggestion. 

According to your suggestion, we revised it in lines 46 and 312.

- l. 68: It should be made clear that reference 9 was written by the same authors based on the same dataset, and it should be explained in which aspects this new paper overlaps with the old one and what it adds.

Response: Thanks very much for your suggestion. 

According to your suggestion, we revised it in lines 71-76.

- l. 83 (and other places): "Multivariate" should be replaced by "multivariable".

Response: Thanks very much for your suggestion. 

According to your suggestion, we revised it in lines 91, 92, 142, 148 and 223.

- l. 121-122: I still don't understand why and how patient records were "deidentified" given that they were anonymized anyway. Can this sentence be omitted? Otherwise pleas add some details.

Response: Thanks very much for your suggestion. 

According to your suggestion, we revised it in lines 128.

- l. 131: Suggest to write "Prevalence and PMR of specific CHDs...".

Response: Thanks very much for your suggestion. 

According to your suggestion, we revised it in lines 137.

- l. 149: Which Government? Suggest to write: "The Hunan Provincial Health Commission routinely collected surveillance data based on the Hunan Maternal and Child Health Surveillance Manual developed by...".

Response: Thanks very much for your suggestion. 

According to your suggestion, we revised it in lines 124-125.

- l. 190-195: Suggest to omit the first two sentences and just write: "Starting with all variables from table 3, the following variables were selected for the final model in stepwise logistic regression analysis: ..."

Response: Thanks very much for your suggestion. 

According to your suggestion, we revised it in lines 197-199.

- l. 196: In order to make the difference to the previous analysis clearer, the sentence might rather begin with "In the logistic regression model with mutual adjustment, perinatal deaths..."

Response: Thanks very much for your suggestion. 

According to your suggestion, we revised it in lines 199-200.

- l. 217: It is strange to call this a meaningful finding given that more or less the same value has been mentioned as result of reference 9 (see also my comment above). The authors should be very clear in distinguishing new and old findings on these data in order to avoid self-plagiarism.

Response: Thanks very much for your suggestion. 

According to your suggestion, we revised it in lines 224, and 71-76.

- It appears dubious that the results and discussion on selective TOPs have been deleted. I understand that selective TOPs are a sensitive topic in China, but the findings cannot be seriously interpreted without explaning this issue properly. In particular, the paragraph in line 217-229 speculates about potential reasons why the PMR of CHDs is so much higher in Hunan (and Zheijang) than in Europe, but these rates just seem to match the rates of selective TOPs. It would e.g. be relevant to mention the PMR for CHDs without selective TOPs and to discuss whether this is comparable to other countries. Further, it should be discussed why there were so much more selective TOPs in Hunan than in Europe, and whether these are more common in rural than urban areas (and if so, why).

Response: Thanks very much for your suggestion. 

According to your suggestion, we revised it in lines 157-158, 228-230, and 251-253.

- I was asking for setting up an online repository which covers both the analysis code and an accompanying data dictionary, see e.g. here: https://osf.io/qg4u6/

The respective URL should be mentioned in the Methods section.

Response: Thanks very much for your suggestion. 

According to your suggestion, we revised it in lines 324-326.

---

## [Editor Report · Decision Letter 3]

15 May 2024

Perinatal deaths attributable to congenital heart defects in Hunan Province, China, 2016-2020

PONE-D-23-39650R3

Dear Dr. Zhou,

We’re pleased to inform you that your manuscript has been judged scientifically suitable for publication and will be formally accepted for publication once it meets all outstanding technical requirements.

Kind regards,

Andreas Beyerlein

Academic Editor

PLOS ONE
---

## [Editor Report · Acceptance letter]

5 Jun 2024

PONE-D-23-39650R3 

PLOS ONE

Dear Dr. Zhou, 

I'm pleased to inform you that your manuscript has been deemed suitable for publication in PLOS ONE. Congratulations! Your manuscript is now being handed over to our production team.

Kind regards, 

on behalf of

Dr. Andreas Beyerlein 

Academic Editor

PLOS ONE